# Occurrence of *Campylobacter* spp. and Phenotypic Antimicrobial Resistance Profiles of *Campylobacter jejuni* in Slaughtered Broiler Chickens in North-Western Romania

**DOI:** 10.3390/antibiotics11121713

**Published:** 2022-11-28

**Authors:** Sebastian Alexandru Popa, Adriana Morar, Alexandra Ban-Cucerzan, Emil Tîrziu, Viorel Herman, Khalid Ibrahim Sallam, Doru Morar, Ulaș Acaroz, Mirela Imre, Tijana Florea, Hamid Mukhtar, Kálmán Imre

**Affiliations:** 1Department of Animal Production and Veterinary Public Health, Faculty of Veterinary Medicine, University of Life Sciences “King Mihai I” from Timişoara, 300645 Timișoara, Romania; 2Department of Infectious Diseases and Preventive Medicine, Faculty of Veterinary Medicine, University of Life Sciences “King Mihai I” from Timişoara, 300645 Timisoara, Romania; 3Department of Food Hygiene and Control, Faculty of Veterinary Medicine, Mansura University, Mansura 35516, Egypt; 4Department of Internal Medicine, Faculty of Veterinary Medicine, University of Life Sciences “King Mihai I” from Timişoara, 300645 Timisoara, Romania; 5Department of Food Hygiene and Technology, Faculty of Veterinary Medicine, Afyon Kocatepe University, Afyonkarahisar 03200, Turkey; 6Department of Parasitology and Dermatology, Faculty of Veterinary Medicine, University of Life Sciences “King Mihai I” from Timişoara, 300645 Timișoara, Romania; 7Institute of Industrial Biotechnology, Government College University, Lahore 54000, Pakistan

**Keywords:** *Campylobacter*, broiler, occurrence, antimicrobial resistance

## Abstract

Campylobacteriosis is recognized as one of the most common food-borne zoonoses, with worldwide distribution, having undercooked poultry meat and other cross-contaminated foodstuffs as the main sources of human infections. The current study aimed to provide data on the occurrence of the thermophilic *Campylobacter* spp. in seven broiler chicken flocks, from three north-western Transylvanian counties of Romania, as well as to determine the antimicrobial resistance profile of the isolated *C. jejuni* strains. A total of 324 fresh cecal samples were collected during the slaughtering process, and screened for the presence of *Campylobacter* spp., using routine microbiological and molecular diagnostic tools. Overall, 85.2% (276/324; 95% CI 80.9–88.6) of the tested samples expressed positive results for *Campylobacter* spp., with dominant occurrence of *C. coli* towards *C. jejuni* (63.4% vs. 36.6%). From the six tested antimicrobials, the 101 isolated *C. jejuni* strains were resistant against ciprofloxacin (79.2%), nalidixic acid (78.2%), tetracycline (49.5%), and streptomycin (7.9%), but total susceptibility was noticed against erythromycin and gentamicin. Seven (6.9%) isolates exhibited multidrug resistance. The study results emphasize the role of broiler chicken as reservoir of *Campylobacter* infections for humans, as well as strengthen the necessity of the prudent using of antimicrobials in the poultry industry.

## 1. Introduction

Out of the 31 species in the genus *Campylobacter* [1], *Campylobacter jejuni* and *Campylobacter coli* are the most frequently (~95%) incriminated in human campylobacteriosis cases [2]. Campylobacteria are Gram-negative, non-spore forming, microaerophilic and oxidase-positive microorganisms, with a spiral or slightly curved form, being very sensitive to environmental factors [1,3]. Its main reservoir is the intestinal tract of wild and domesticated birds and mammals [4].

*Campylobacter* transmission to humans can occurs through direct contact with farm animals or by consumption of contaminated water and/or animal origin foodstuffs (e.g., untreated drinking water, unpasteurized milk, different types of meat) [5,6,7,8]. Out of those factors, raw or undercooked chicken meat is the biggest potential source of *Campylobacter* during human consumption. The contamination of this type of meat usually occurs during different stages of the slaughtering process (e.g., evisceration, cleaning, or chilling) [9,10,11,12,13]. Campylobacteriosis symptoms, as human bacterial gastroenteritis, include diarrhea, acute abdominal pain, cramps, vomiting, and fever. In addition, some patients can develop severe systemic and neurological disorders, such as hemolytic-uremic or Guillain–Barré syndromes [14,15,16]

In recent years, bacterial foodborne pathogen resistance to antimicrobials has gradually increased, as a consequence of the large-scale and imprudent usage of antimicrobials in different industry categories (e.g., medical, livestock, poultry, agriculture), and has become a serious global public health concern [17,18,19]. In particular, antimicrobial resistance (AMR) of animal and human-origin *Campylobacter* strains have also increased [20]. In this regard, high resistance rates have been reported to tetracyclines and quinolones [21], and in the last few years, also towards aminoglycosides [22]. The most recently published open access summarized report by the European Food Safety Authority (EFSA) and European Centre for Disease Prevention and Control (ECDC), on the frequency of isolation and antimicrobial susceptibility profile of food-borne pathogens, underlined a worryingly high antimicrobial resistance profile of the tested *Campylobacter* spp. strains isolated from broiler meat samples towards tetracycline (57.6–67.7%), ciprofloxacin (64.2–90.0%), and nalidixic acid (71.0–84.8%) [22]. Furthermore, the isolation of multi-drug resistance (MDR) *Campylobacter* spp. strains has considerably increased [21,23]. In addition, these results strengthen the necessity and importance of the using of a standardized and continuous AMR surveillance system in food animals, especially at the level of slaughtering and food processing units, to prevent the further dispersion and transmission of MDR strains. In healthy domestic poultry, *Campylobacter* is considered an indicator commensal bacterium, frequently used for the monitoring of AMR [24]. Likewise, this bacteria has the capacity to acquire AMR faster than other commonly isolated bacteria from different animal-origin foodstuffs [18].

Given this context, to date, limited information is available concerning the monitoring of the occurrence and antimicrobial susceptibility profile of poultry origin *Campylobacter* spp. in Romania [25,26]. Therefore, the present study aimed to provide data on the isolation frequency of *Campylobacter* spp., and the phenotypic antimicrobial resistance profile of *C. jejuni*, in cecal samples of healthy slaughtered broiler chicken, in different abattoirs, of three north-western Transylvanian counties of Romania.

## 2. Results

### 2.1. Prevalence of Campylobacter spp.

The distribution of *Campylobacter* spp. positive findings in the tested broiler chicken cecum samples, according to their provenience and study years, are presented in Table 1.

Overall, 85.2% (276/324; 95% CI 80.9–88.6) of the tested samples expressed positive results, with relatively uniform distribution within the study years, with the mention that only three farms (A, F, and G) were monitored in each of the study years. The prevalence of *Campylobacter* spp. in the investigated farms largely varied from 53.3% (farm D) to 100% (farm B).

Results of routine microbiological investigations in the isolation of *Campylobacter* spp. were molecularly confirmed for all the isolated strains. The species-specific PCR reaction results showed that 175 (63.4%; 95% CI 57.6–68.9) isolates were *C. coli*, and 101 (36.6%; 95% CI 31.1–42.4) were *C. jejuni*. In all the investigated farms, *C. coli* expressed dominant occurrence (Table 1).

### 2.2. Antimicrobial Resistance Profiles of the Campylobacter jejuni Isolates

Antimicrobial susceptibility testing of 101 *C. jejuni* isolates expressed resistance in descending order towards ciprofloxacin (CIP) (79.2%), nalidixic acid (NA) (78.2%), tetracycline (TET) (49.5%), and streptomycin (STM) (7.9%) (Table 2). All the isolated strains were susceptible to erythromycin (ERY) and gentamycin (GEN). Comparison of the expressed AMR profile results by the isolated *C. jejuni* strains towards the tested drugs, between the studied years indicated a constant increased trend in the case of NA from 2016 to 2020, and a higher resistance rate in 2018 for CIP, TET, and STM compared with the years of 2016 and 2020.

For thirty-two isolates (31.7%; 95% CI 23.4–41.3), resistance was noticed against antimicrobials from the quinolones group (CIP + NA), with resistance percentages of 37.7%, 17.4%, and 29.4% in 2016, 2018, and 2020, respectively (Table 3). Another 41 strains (40.6%; 95% CI 31.5%–50.3) were also accounted as low drug resistance, showing resistance against drugs from the quinolones (CIP and NA), and tetracycline (TET) classes. It is noteworthy that seven (6.9%; 95% CI 3.4–13.6) isolates expressed resistance towards more than three antimicrobials, from different classes, namely quinolones (CIP and NA), tetracyclines (TET), and aminoglycosides (STM). They were classified as MDR [21], with an occurrence, in each of the studies years, in three out of seven investigated farms (Table 3).

## 3. Discussions

This is the first published study in Romania providing information in the occurrence of *Campylobacter* spp. in slaughter-age broiler chickens processing cecal matrices. The recorded high overall prevalence value (85.2%) of *Campylobacter* spp. in the screened broiler chickens suggests their potential reservoir role for human infections. Several studies, conducted at a worldwide level, have investigated the occurrence of *Campylobacter* spp. in broiler cecal content, highlighting different frequency of isolation rates. Thus, higher values of isolations have been recorded in the Santa Catarina state of Brazil (100%) [27] and the Sharkia Governatore of Egypt (88.6%) [28], but lower in the Pichincha province of Ecuador (69.6%) [29], North Lebanon (67.0%) [30], five provinces of Sri Lanka (63.8%) [31], and the north-east of Tunisia (22.4%) [32]. Considering other available European studies, investigating the occurrence of the commensal *Campylobacter* spp. in the same matrices of broiler chickens, lower prevalence values have been reported in Italy (78.8%) [33], Greece (73.9%) [34], Hungary (60.1%) [35], and Spain (38.1%) [36]. Variations in the prevalence of *Campylobacter* spp. among different investigations can be related by several factors, including sample size and sampling methodology, the applied isolation and identification techniques, and the monitored geographical regions or environmental conditions. Likewise, in the present study, large variations in pathogen detection has been recorded between the investigated farms (ranging from 53.3% to 100%), explainable by the implementation of different biosecurity measures and management practices to monitor and control microbial infections.

In the present investigation, *C. coli* expressed dominant occurrence towards *C. jejuni* (63.4% vs. 36.6%), with both species having a potential public health threat. As a term of comparison, several studies have demonstrated the same pattern (44.1% vs. 39.5% in Lebanon [30]; 68.7% vs. 18.9% in Ecuador [29]; and 70.6 vs. 29.4 in Italy [37]), while others indicated the dominance of *C. jejuni* (100% vs. 70.6% in Brazil [27]; 77.4% vs. 22.6% in Egypt [28]; 55.5% vs. 48.8% in Hungary [35]; and 68.9% vs. 31.1% in Tunisia, [32]) in broiler origin samples, but without a clear scientific explanation supporting the irregular variability of these species from one country or region to another.

In the present study, analysis of AMR data of the tested *C. jejuni* isolates, in the present study, indicated resistance towards four out of six of the tested antimicrobials, namely CIP (79.2%), NA (78.2%), TET (49.5%), and STM (7.9%) (Table 2), either alone or under different combination forms (Table 3). These values of resistance are in line with those reported for broiler-origin *C. jejuni* strains in the most recently published harmonised epidemiological cut-off value (ECOFF) EFSA and ECDC summarizing report, from the level of EU countries (CIP–72.8%; NA–69.2%; TET–52.7%; STM–15.6%), as well as with the data provided by Romania (CIP–82.0%; NA–80.4%; TET–57.4%; STM–9.9%) (22). As has been previously highlighted by [38,39,40], it is noteworthy that all strains were susceptible against ERY and GEN, even if Romania was accounted as contributing country (ERY–2.2%; GEN–1.2%) in 2020 to the recorded low resistance rate to these drugs at the EU countries level (ERY–0.8%; GEN–0.1%). Other studies monitoring the antimicrobial susceptibility profile of broiler-origin *C. jejuni* isolates, conducted in different slaughtering units of the Transylvania region of Romania, indicated different resistance patterns for CIP (100%, [26]; 9.1%, [25]), NA (100% [26]; 9.1% [25]), and TET (100%, [26]; 31.8%, [25]), and total susceptibility for STM, ERY and GEN (0.0% [25,26]). The recorded differences between these results can constitute an important indicator of antimicrobial use, as well as AMR in Romanian broiler farms. In addition, different representative investigations, conducted at the worldwide level, underscored a wide variable resistance spectrum of the tested *C. jejuni* strains for CIP (98.9% in Tunisia [32]; 97.9% in Ecuador [29]; 58.7% in Lithuania [39]; 42.4% [41] and 39.0% [37] in Italy), NA (100% in Ecuador [29]; 77.8% in Hungary [35]; 60.3% in Lithuania [39]; 57.1% in Tunisia [32]; and 45.3% [41] and 39.0% [37] in Italy), TET (100% in Tunisia [32]; 83.3% in Ecuador [29]; 33% in Hungary [35]; 26.4% in Lithuania [39]; 25.0% [41] and 22.2% in Bulgaria [42]; and 10% [37] in Italy), STM (29.0% [37] and 1.6% [41] in Italy; 8.3% in Sri Lanka [31]; 5.9% in Lithuania [39]), ERY (100% in Tunisia [32]; 4.4% in Bulgaria [42]; 4.2% in Ecuador [29]; 3.1% in Italy [41]; 1.6% in Hungary [35]), and GEN (14.3% in Tunisia [32]; 1.6% in Italy [41]).

An overview of the recorded great variations in AMR of *C. jejuni* strains in these studies, from extremely high (especially in case of quinolones and tetracyclines), to relatively low (aminoglycosides), can be accounted for by different levels of uncontrolled usage of these antimicrobials in veterinary medicine with therapeutic or prophylactic purposes. This idea is reinforced by the recorded constant increased resistance trend for NA from 2016 to 2020.

Of note, we found seven (6.9%) MDR *C. jejuni* isolates. This value is much lower than the ones previously published in Romania (100%, [26]; 40.9%, [25]) and, in general, in other studies reporting the worrisome MDR phenomenon for broiler-origin *C. jejuni* strains in Algeria (100%, [43]); Portugal (74.4%, [44]) and Tunisia (100% [32]). These findings are linked by the unrestricted and preferred usage of some antimicrobials, without any prescription, in the poultry industry, highlighting the urgent need to control antimicrobials usage in animal production.

The recorded total susceptibility against macrolides and some aminoglycosides of the tested *C. jejuni* strains in the present study suggests that ERY and GEN can still be used as promising tools for the management of human campylobacteriosis cases. Among the antimicrobial classes tested in the present investigation, macrolides are considered the first line drugs for *Campylobacter* infections treatment, beside quinolones and aminoglycosides, which has a lesser extent, whereas tetracyclines constitute and alternative option [45].

## 4. Materials and Methods

### 4.1. Sample Collection

The study was undertaken in the years 2016, 2018 and 2020, respectively. A total of 324 randomly selected broiler chicken cecal samples were collected during the slaughtering process in seven slaughterhouses (A, B, C, D, E, F, G), located in three north-western Transylvanian counties (Maramureș, Satu-Mare and Cluj) of Romania. Thus, 180 samples were collected in 2016, 89 samples in 2018, and 55 samples in 2020. The samples were collected monthly, within the national AMR monitoring program of indicator poultry for commensal and zoonotic bacteria, in accordance with the EU Decision No. 652/2013 [46]. The study included birds between 36 and 51 days of life, free of antimicrobial treatments, with different farm origins, in each of the enrolled slaughterhouses. All the investigated broiler chicken farms were designed for intensive production. The samples were placed in sterile plastic containers, stored, and transported under refrigeration (~4 °C) conditions, in an isothermal box, to the microbiology laboratory of a sanitary veterinary directorate of a north-western county. The samples were submitted for microbiological examination within 24 h of their arrival to the laboratory.

### 4.2. Isolation, and Molecular Identification of Campylobacter spp.

The detection and identification of *Campylobacter* spp. commensal strains were performed by direct plating from broiler cecum samples, according to the ISO 10272-1:2017 standard [47]. In brief, a small incision was made at the cecum level, and a loopful (10 µL) of the content was streaked onto the first half of Butzler (Oxoid Ltd., Basingstoke, UK), and Charcoal Cefoperazone Deoycholate (mCCDA) (Oxoid Ltd.) selective agars. Next, a second sterile plastic loop was used to inoculate the second half of the plates in the same way. The *C. jejuni* ATCC 33291 and *C. coli* ATCC 43478 served as positive control reference strains, to ensure the specific and relevant results. The plates were incubated at 41.5 °C, for 44 h, in microaerobic environment (5% O_2_, 10% CO_2_, 85% N_2_) in jars, using microaerobic bags (Thermo Scientific^TM^, Waltham, MA, USA), and then examined. Approximately four typical suspected colonies of presumptive *Campylobacter* spp. were selected and picked up onto a non-selective Columbia blood agar plate (Oxoid Ltd.,), and then incubated for 44 h, at 41.5 °C, in microaerobic conditions in order to obtain pure colonies.

After the incubation period, the morphology and motility of the freshly grown pure colonies, together with their growing capacity at 25 °C, were studied. In addition, the biochemical characterization of the presumed *Campylobacter* isolates, including the catalase and oxidase tests (oxidase detection strips, Oxoid Ltd.) were performed. Isolates, identified as *Campylobacter* spp., were subsequently submitted to the Institute for Hygiene and Public Health, București, Romania for confirmation of *Campylobacter* genus, and differentiation at species level, by using a simplex polymerase chain reaction (PCR). The genus and species-specific primers set and cycling conditions were used as previously designed by Linton et al. [48].

### 4.3. Antimicrobial Susceptibility Tests of C. jejuni

Following the EU Decision No. 652/2013 recommendation [46], only the isolated *C. jejuni* (n = 101) strains were tested using the microdilution (MIC) technique. The grown colonies on Columbia blood agar (Oxoid Ltd.) were prepared in Tryptic soy broth, and adjusted to a turbidity of 0.5 McFarland standard. The resulted mixture was seeded in Müeller–Hinton broth (Oxoid Ltd.), supplemented with 2.5–5% lysed horse blood, and subsequently distributed into EUCAMP microtitre plates with concentrations of the following antimicrobials: ERY (1–128 µg/mL), CIP (0.125–16 µg/mL), TET (0.5–64 µg/mL), GEN (0.125–16 µg/mL), NA (1–64 µg/mL), and STM (0.25–16 µg/mL). After inoculation, the microplates were incubated at 37 °C, in a microaerobic atmosphere, for 48 h. The results were interpretated according to EUCAST breakpoints (European Committee on Antimicrobial Susceptibility Testing) [49].

## 5. Conclusions

This study demonstrated a high frequency of *Campylobacter* spp. isolation (85.2%) from the tested broiler chicken cecum samples, confirming their reservoir status for human campylobacteriosis. In addition, our results provided evidence that the AMR phenomenon is common among the isolated *C. jejuni* strains, particularly towards quinolones and tetracyclines. Likewise, the emergence of MDR strains reinforces the necessity to reduce the routine use of antimicrobials in the poultry industry, or to replace them with medicinal plant-derived alternative drugs. Even if the provided phenotypic AMR data of the present investigation can be considered suggestive, further studies, focusing on the molecular evidence of AMR and virulence genes, beside the molecular typing of phenotypically similar *Campylobacter* isolates, along the entire food chain, are still necessary, in order to improve the necessary knowledge for control of human campylobacteriosis in our country.

## Figures and Tables

**Table 1 antibiotics-11-01713-t001:** Distribution of the identified *Campylobacter* spp. in the tested broiler chicken cecum samples, according to their origin and study year.

Farm/Slaughterhouse	*Campylobacter* spp.	Study Years			Total (%)
		2016	2018	2020	
		Positive/Total Investigated (%)			
A	*C. coli*	11/30 (36.7)	6/9 (66.7)	9/13 (69.2)	26/52 (50.0)
	*C. jejuni*	13/30 (43.3)	1/9 (11.1)	4/13 (30.8)	18/52 (34.6)
	Neg.	6/30 (20.0)	2/9 (22.2)	0/13 (0.0)	8/52 (15.4)
B	*C. coli*	8/10 (80.0)	N.A.	N.A.	8/10 (80.0)
	*C. jejuni*	2/10 (20.0)	N.A.	N.A.	2/10 (20.0)
	Neg.	0/10 (0.0)	N.A.	N.A.	0/10 (0.0)
C	*C. coli*	15/20 (75.0)	N.A.	N.A.	15/20 (75.0)
	*C. jejuni*	3/20 (15.0)	N.A.	N.A.	3/20 (15.0)
	Neg.	2/20 (10.0)	N.A.	N.A.	2/20 (10.0)
D	*C. coli*	9/30 (30.0)	N.A.	N.A.	9/30 (30.0)
	*C. jejuni*	7/30 (23.3)	N.A.	N.A.	7/30 (23.3)
	Neg.	14/30 (46.6)	N.A.	N.A.	14/30 (46.7)
E	*C. coli*	12/30 (40.0)	15/18 (83.3)	N.A.	27/48 (56.2)
	*C. jejuni*	15/30 (50.0)	3/18 (16.7)	N.A.	18/48 (37.5)
	Neg.	3/30 (10.0)	0/18 (0.0)	N.A.	3/48 (6.3)
F	*C. coli*	15/30 (50.0)	18/29 (62.0)	11/22 (50.0)	44/70 (62.8)
	*C. jejuni*	10/30 (33.3)	9/29 (31.1)	5/22 (22.7)	24/70 (34.3)
	Neg.	5/30 (16.7)	2/29 (6.9)	6/22 (27.3)	13/70 (18.4)
G	*C. coli*	15/30 (50.0)	20/33 (60.6)	11/20 (55.0)	46/83 (55.4)
	*C. jejuni*	11/30 (36.7)	10/33 (30.3)	8/20 (40.0)	29/83 (34.9)
	Neg.	4/30 (13.3)	3/33 (9.1)	1/20 (5.0)	8/83 (9.7)
Total	n.a.	146/180 (81.1)	82/89 (92.1)	48/55 (87.3)	n.a.

Legend: Neg.= negative; N.A. = not available; n.a. = not applicable.

**Table 2 antibiotics-11-01713-t002:** Antimicrobial resistance of *Campylobacter jejuni* strains.

Antimicrobials	Cut-Off Values	MIC Breackpoint (µg/mL)		No. of Resistant /Total Investigated (%) *C. jejuni* Strains in the Study Years			Total (%)
		S	R	2016	2018	2020	
Erythromycin	4	1	128	0/61 (0.0)	0/23 (0.0)	0/17 (0.0)	0 (0.0)
Ciprofloxacin	0.5	0.125	16	42/61 (68.9)	23/23 (100)	15/17 (88.2)	80 (79.2)
Tetracycline	1	0.5	64	24/61 (39.3)	18/23 (78.3)	10/17 (58.8)	52 (49.5)
Gentamicin	2	0.125	16	0/61 (0.0)	0/23 (0.0)	0/17 (0.0)	0 (0.0)
Nalidixic acid	16	1	64	40/61 (65.6)	22/23 (95.7)	17/17 (100)	79 (78.2)
Streptomycin	4	0.25	16	4/61 (6.6)	3/23 (13.0)	1/17 (5.9)	8 (7.9)

Legend: S = susceptible; R = resistant.

**Table 3 antibiotics-11-01713-t003:** Antimicrobial resistance profile of the isolated *Campylocater jejuni* strains (n = 101).

Farm/ Slaughterhouse	Resistance to Antimicrobial Profile of *Campylobacter jejuni* Isolates (%) during the Study Years								
	2016			2018			2020		
	CIP+NA	CIP+TET+ NA	CIP+TET+NA+ STM	CIP+NA	CIP+TET+ NA	CIP+TET+NA+ STM	CIP+NA	CIP+TET+ NA	CIP+TET+NA+ STM
A	10/13 (76.9)	0/13 (0.0)	0/13 (0.0)	1/1 (100)	0/1 (0.0)	0/1 (0.0)	3/4 (75.0)	0/4 (0.0)	0/4 (0.0)
B	2/2 (100)	0/2 (0.0)	0/2 (0.0)	N.A.			N.A.		
C	2/3 (66.7)	0/3 (0.0)	0/3 (0.0)	N.A.			N.A.		
D	0/7 (0.0)	0/7 (0)	3/7 (42.8)	N.A.			N.A.		
E	3/15 (20.0)	7/15 (46.7)	0/15 (0.0)	0/3 (0.0)	2/3 (66.6)	1/3 (33.3)	N.A.		
F	0/10 (0.0)	5/10 (50.0)	0/10 (0.0)	1/9 (11.1)	6/9 (66.6)	1/9 (11.1)	1/5 (20.0)	4/5 (80.0)	0/5 (0.0)
G	5/11 (45.4)	3/11 (27.3)	0/11 (0.0)	2/10 (20.0)	8/10 (80.0)	1/10 (10.0)	1/8 (12.5)	6/8 (75.0)	1/8 (12.5)
Total	23/61 (37.7)	15/61 (24.6)	3/61 (4.9)	4/23 (17.4)	16/23 (69.6)	3/23 (13.1)	5/17 (29.4)	10/17 (58.8)	1/17 (5.9)

Legend: N.A.= not available, CIP—ciprofloxacin, NA—nalidixic acid, TET—tetracycline, STM—streptomycin.

## Data Availability

Not applicable.

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
