# Peer review of "Occurrence of Campylobacter spp. and Phenotypic Antimicrobial Resistance Profiles of Campylobacter jejuni in Slaughtered Broiler Chickens in North-Western Romania"

_antibiotics, 2022, doi:10.3390/antibiotics11121713_

Round 1

Reviewer 1 Report

Dear authors 

Together with minor editing listed below, the comparison of the antibiotic-resistant profile results between the studied years could be a good addition to establish if potential increases or decreases in antibiotic susceptibilities happened.

 L 2 and 3 Please consider the use of another title. An option could be "Occurrence of Campylobacter spp. and antimicrobial resistance profiles of Campylobacter jejuni in slaughtered broiler chickens in North-Western Romania"

L33 Please clarify what are you referring to when use the term "classical".

L45 Please use the full name of both species of Campylobacter

L53 Please highlight that raw or undercooked chicken meat is the biggest potential source of Campylobacter during human consumption

L62 Please use a term more specific than "...different type of sectors" 

L73 Please add "also" at the end of the sentence after "increased".

L77 Please add a comma after "In healthy poultry"

L102 Please use "isolation" instead of "detection"

L 107, 112, 130, and 131 Please the full name of this bacterial species

L132 to 134 Please include the antibiotic names for each of abbreviated names included in the subheading on Table 3 

L137 Please consider deleting the sentence "To the best author´s knowledge"

L138 and 139 Please delete ", at the level of conventional broiler flock,"

L 143 Please use "isolation" instead of "isolations"

L143 Please change from "Thus, higher value has..." to "Thus, higher values of isolations have been..."

L149 Please use the plural of region

L154 Please add a comma after "In the present investigation"

L 155 Please replace the term "a public health importance" with "a potential public health threat"

L 161 Please begin the sentence with "In the present study,"

L169 Please delete "by several scientific publications"

L174 What are you referring to as "historical"? If the term is not related to the study you performed, please delete it.

L189 After "Of note" please use "we found 7 (6.9%) MDR C. jejuni isolates"

L190 Please use "the one" instead of "have been"

L205 Please clarify the years in which this study was performed including 2016, 2018, and 2020. 

L 224, 233, 238, 247, and 249 Please delete "Basingstoke, UK" in order to avoid repeating information.

L260 and 261 Please avoid repeating results. Please delete "with dominant occurrence of C. coli towards C. jejuni (63.4% vs. 36.6%)".

L 261 and 270 Please use the term "human campylobacteriosis" or "human Campylobacter spp. infections" instead of "human campylobacter infections"

L262 Please use "... our results provided ..." instead of  "... the results provide ..."

Author Response

Reviewer #1

Dear authors,

Together with minor editing listed below, the comparison of the antibiotic-resistant profile results between the studied years could be a good addition to establish if potential increases or decreases in antibiotic susceptibilities happened.

Dear reviewer, thank you very much for your overall positive feedback about the quality of our submission, and your valuable comments helping us to improve the quality of the manuscript.

According to the reviewer suggestion, the following sentence was inserted in the Results section of the revised version of the manuscript: “Comparison of the expressed AMR profile results by the isolated C. jejuni strains towards the tested drugs, between the studied years, indicated a constant increased trend in case of NA from 2016 to 2020, and a higher resistance rate in 2018 for CIP, TET and STM compared with the years of 2016 and 2020.” And in the Discussion section “… can be accounted by different levels of uncontrolled usage of these antimicrobials, in veterinary medicine, with therapeutic or prophylactic purposes. This idea, is reinforced by the recorded constant increased resistance trend for NA from 2016 to 2020.”

L 2 and 3 Please consider the use of another title. An option could be "Occurrence of Campylobacter spp. and antimicrobial resistance profiles of Campylobacter jejuni in slaughtered broiler chickens in North-Western Romania"

Special thanks for your suggestion. The title was changed accordingly, resulting in „Occurrence of Campylobacter spp. and phenotypic antimicrobial resistance profiles of Campylobacter jejuni in slaughtered broiler chickens in North-Western Romania”

L33 Please clarify what are you referring to when use the term "classical".

The term classical referred to the most frequently used microbiological testing methods. In order to avoid any confusion, the term „classical” was replaced with „routine microbiological”.

L45 Please use the full name of both species of Campylobacter

The correction was done.

L53 Please highlight that raw or undercooked chicken meat is the biggest potential source of Campylobacter during human consumption

The suggested additional information was inserted in the revised version.

L62 Please use a term more specific than "...different type of sectors"

According to the reviewer suggestion the term "...different type of sectors" was replaced with „industry categories”

L73 Please add "also" at the end of the sentence after "increased".

The term „also” was added according to the reviewer suggestion.

L77 Please add a comma after "In healthy poultry"

The comma was inserted as requested.

L102 Please use "isolation" instead of "detection"

Replaces as requested.

L 107, 112, 130, and 131 Please the full name of this bacterial species

As requested, the full name for Campylobacter species was used.

L132 to 134 Please include the antibiotic names for each of abbreviated names included in the subheading on Table 3

Was included according to the reviewer suggestion.

L137 Please consider deleting the sentence "To the best author´s knowledge"

According to the reviewer suggestion the formulation "To the best author´s knowledge" was deleted

L138 and 139 Please delete ", at the level of conventional broiler flock,"

Deleted according to the reviewer request.

L 143 Please use "isolation" instead of "isolations"

Changed as requested.

L143 Please change from "Thus, higher value has..." to "Thus, higher values of isolations have been..."

Changed according to the reviewer suggestion.

L149 Please use the plural of region

Used, as requested.

L154 Please add a comma after "In the present investigation"

Added, as requested.

L 155 Please replace the term "a public health importance" with "a potential public health threat"

Was replaces according to the reviewer recommendation.

L 161 Please begin the sentence with "In the present study,"

The modification was done as requested.

L169 Please delete "by several scientific publications"

Deleted as requested.

L174 What are you referring to as "historical"? If the term is not related to the study you performed, please delete it.

Deleted, as reqested.

L189 After "Of note" please use "we found 7 (6.9%) MDR C. jejuni isolates"

The sentence was modified according to the reviewer suggestion.

L190 Please use "the one" instead of "have been"

Changed according to the reviewer request.

L205 Please clarify the years in which this study was performed including 2016, 2018, and 2020.

Was clarified as the reviewer requested.

L 224, 233, 238, 247, and 249 Please delete "Basingstoke, UK" in order to avoid repeating information.

The "Basingstoke, UK" was deleted according to the reviewer request.

L260 and 261 Please avoid repeating results. Please delete "with dominant occurrence of C. coli towards C. jejuni (63.4% vs. 36.6%)".

Deleted, as the respected reviewer requested.

L 261 and 270 Please use the term "human campylobacteriosis" or "human Campylobacter spp. infections" instead of "human campylobacter infections"

Replaced in accordance with the reviewer suggestion.

L262 Please use "... our results provided ..." instead of "... the results provide ..."

Replaced as suggested.

Thank you again for your time and valuable comments!

Reviewer 2 Report

Dear authors

Thanks for your effort in the work and for your presentation of it.

However, some comments could be taken in consideration during the revision of the article.

1. Were there any signs in chickens during the rearing in the farm that gave an indication of Campylobacter suspicion?

2. What about detection of the resistance genes of the isolated Campylobacter species?!

3. What about the detection of antimicrobial resistance genes to complete the picture of antibiotic resistance of the isolated Campylobacter species.

Best wishes

Author Response

Reviewer #2

Dear authors,

Thanks for your effort in the work and for your presentation of it.

Dear Respected Reviewer,

The authors would like to express their gratitude towards your overall positive feedback regarding the quality of the present submission!

However, some comments could be taken in consideration during the revision of the article.

  1. Were there any signs in chickens during the rearing in the farm that gave an indication of Campylobacter suspicion?

No, no any indicativ clinical sign for campylobacteriosis have been recorded during the birds rearing. This aspect have been highlighted within the submitted original version of the manuscript within the line 85, mentioning that “… samples of healthy slaughtered broiler chicken,...”

  1. What about detection of the resistance genes of the isolated Campylobacter species?!
  2. What about the detection of antimicrobial resistance genes to complete the picture of antibiotic resistance of the isolated Campylobacter species.

The authors acknowledged that it would have been better that all isolates had been tested for the presence of antimicrobial resistance genes, but the existent financial resources for the present study were very limited. However, this concern have been highlighted as study limitation and future perspective within the conclusion section of the original submission “Even if the provided phenotypic AMR data of the present investigation can be considered suggestive, further studies, focusing on the molecular evidence of AMR and virulence genes, beside the molecular typing of phenotypically similar Campylobacter isolates, along the entire food chain, are still necessary, in order to improve the necessary knowledge for control of human campylobacteriosis in our country.” Moreover, the new version of the title, is more appropriate on the study aim, highlighting that the present study focused on the evidence of the phenotypic AMR profiles of C. jejuni isolates.

Best wishes,

Thank you again for your time and suggestions!

Reviewer 3 Report

The authors present study on occurrence and antimicrobial resistance of Campylobacter species in broilers in NW Romania. Although the results are of limited scientific importance and novelty, their data helps to complete general knowledge of the most commonly reported gastrointestinal bacterial pathogen in the world.

Title

Title is bit misleading as I was expecting that the AMR was done on all isolates, not just C. jejuni. I consider it should be pointed out in the title.

Introduction

Line 72: I believe this is wrong citation, and that number 22 should be here instead. Please check all literature

Results

Line 109: Names of antimicrobials should be written in full first time when they appear in the text, and this is the place. Materials and methods come later in concept of MDPI manuscripts so it should be done here, and not in M&M.

Line 109 and 122: Erase “respectively”, wrong use.

Line 119: Word “unfortunately” is not a part of scientific language, please, find another word.

Discussions

Line 139-141: Sentence should be rephrased, although this is probably true, without any genotyping data authors cannot claim this from their results.

Line 143-146: I would expect comparison with closer countries, like Bulgaria, Serbia, Ukraine, Hungary, or other countries in Europe.

Lines 156 - 158, 175 - 176 and especially 179 - 184: This is very hard to follow as the literature is presented with numbers. I would advise authors to put names of the countries (and include data for surrounding countries, as stated before). It would be even better to put it in table, than everything would be much easier to follow.

Lines 164-168: There should be a literature cited here. I believe it is number 22.

Lines 167 & 168: I believe that STR is abbreviation for streptomycin and that it should be STM as in whole text.

Line 194: I assume authors meant “antimicrobials” and not “AMR”.

Materials and Methods

Line 229: Please specify the producer of microaerobic bags.

Line 229-234: There is methodological mistake here. I don’t understand what the authors put on the non-selective agar. The colonies that were tested for microscopy and were Gram stained could not be used for this, and if they put other colonies then they should test them after growing on non-selective agar. If fact that is how it should be done; take suspected colonies, streak them on non-selective agar and then test pure colonies that grew on it.

Author Response

Reviewer #3

The authors present study on occurrence and antimicrobial resistance of Campylobacter species in broilers in NW Romania. Although the results are of limited scientific importance and novelty, their data helps to complete general knowledge of the most commonly reported gastrointestinal bacterial pathogen in the world.

Dear Respected Reviewer,

The authors would like to express their gratitude towards your overall positive feedback regarding the quality of the present submission!

Title

Title is bit misleading as I was expecting that the AMR was done on all isolates, not just C. jejuni. I consider it should be pointed out in the title.

In agreement with the reviewer requirement, the title has been rephrased resulting in “Occurrence of Campylobacter spp. and phenotypic antimicrobial resistance profiles of Campylobacter jejuni in slaughtered broiler chickens in North-Western Romania”.

Introduction

Line 72: I believe this is wrong citation, and that number 22 should be here instead. Please check all literature

The authors completely agree the reviewer observation! The mistake was corrected.

Results

Line 109: Names of antimicrobials should be written in full first time when they appear in the text, and this is the place. Materials and methods come later in concept of MDPI manuscripts so it should be done here, and not in M&M.

The authors completely agree the reviewer observation! The mistake was corrected.

Line 109 and 122: Erase “respectively”, wrong use.

In agreement with the reviewer opinion the mistake was corrected.

Line 119: Word “unfortunately” is not a part of scientific language, please, find another word.

The word “unfortunately” was replaced with “it is noteworthy that”

Discussions

Line 139-141: Sentence should be rephrased, although this is probably true, without any genotyping data authors cannot claim this from their results.

The authors agree the reviewer comment. Therefore, the sentence was rephrased resulting in: “The recorded high overall prevalence value (85.2%) of Campylobacter spp. in the screened broiler chickens can suggests their potential reservoir role for human infections”.

Line 143-146: I would expect comparison with closer countries, like Bulgaria, Serbia, Ukraine, Hungary, or other countries in Europe.

According to the reviewer requirement, the following sentence was inserted: „Considering other available European studies, investigating the occurrence of the commensal Campylobacter spp. in the same matrices of broiler chickens, lower prevalence values have been reported in Italy (78.8%) [33], Greece (73.9%) [34], Hungary (60.1%) [35], and Spain (38.1%) [36].”

No available scientific publications in PubMed and WOSCC databases has found investigating broiler cecal content for Campylobacter spp. in Bulgaria, Serbia and Ukraine.

Lines 156 - 158, 175 - 176 and especially 179 - 184: This is very hard to follow as the literature is presented with numbers. I would advise authors to put names of the countries (and include data for surrounding countries, as stated before). It would be even better to put it in table, than everything would be much easier to follow.

The authors chose the first presented option by the reviewer, putting the name of the countries for each of the presented corresponding values.

Please see the lines 150-153, and 187-196 of the revised version.

The referred lines 175-176, of the original submitted version, present data only from 2 different studies conducted in Romania.

Lines 164-168: There should be a literature cited here. I believe it is number 22.

Yes, special thanks Dear Reviewer, we completely agree your suggestion and we remediated the mistake.

Lines 167 & 168: I believe that STR is abbreviation for streptomycin and that it should be STM as in whole text.

According to the reviewer observation, the inappropriate using form of streptomycin abbreviation was corrected.

Line 194: I assume authors meant “antimicrobials” and not “AMR”.

The mistake was corrected, thank you!

Materials and Methods

Line 229: Please specify the producer of microaerobic bags.

The producer of microaerobic bags is Thermo ScientificTM, Waltham, MA, USA

Line 229-234: There is methodological mistake here. I don’t understand what the authors put on the non-selective agar. The colonies that were tested for microscopy and were Gram stained could not be used for this, and if they put other colonies then they should test them after growing on non-selective agar. If fact that is how it should be done; take suspected colonies, streak them on non-selective agar and then test pure colonies that grew on it.

In order to clarify the regrettable occurred confusion, the authors deleted “After the application of Gram staining and morphology examination under light microscopy …” and “(Gram negative and spiral or slightly curved)” formulations from the sentence.

Thank you again for your time and efforts!

Reviewer 4 Report

 The study shows interesting findings and improve the knowledge in the field. The paper deals with the topic of actuality.

 The paper is clear, well written, and the organization is very good.

 The references are up to date, and they are well organized according to the format required by the journal.

Author Response

Reviewer #4

The study shows interesting findings and improve the knowledge in the field. The paper deals with the topic of actuality. The paper is clear, well written, and the organization is very good.The references are up to date, and they are well organized according to the format required by the journal.

Dear Reviewer,

The authors want to express their gratitude towards your overall positive feddback regarding the quality of the present work. Thank you for your time and efforts!

Best wishes,

Dr. Popa,

Round 2

Reviewer 2 Report

The author responded positively all comments and answered them.

In addition, (He/she) made the required modifications in the revised updated version of the manuscript.

Thanks and best wishes.